

# The variability of emotions, physical complaints, intention, and self-efficacy: an ecological momentary assessment study in older adults

Iris Maes[1], Lieze Mertens[1,2,3], Louise Poppe[2,3], Geert Crombez[4], Tomas Vetrovsky[5] and Delfien Van Dyck[1]

[1] Department of Movement and Sports Sciences, Ghent University, Gent, Belgium
[2] Department of Public Health and Primary Care, Ghent University, Gent, Belgium
[3] Fund for Scientific Research Flanders (FWO), Brussels, Belgium
[4] Department of Experimental-clinical and health psychology, Ghent Univeristy, Gent, Belgium
[5] Faculty of Physical Education and Sport, Charles University Prague, Prague, Czech Republic

Corresponding author
Iris Maes, iris.maes@ugent.be

## ABSTRACT

**Background**. Many theoretical frameworks have been used in order to understand health behaviors such as physical activity, sufficient sleep, healthy eating habits, etc. In most research studies, determinants within these frameworks are assessed only once and thus are considered as stable over time, which leads to rather 'static' health behavior change interventions. However, in real-life, individual-level determinants probably vary over time (within days and from day to day), but currently, not much is known about these time-dependent fluctuations in determinants. In order to personalize health behavior change interventions in a more dynamic manner, such information is urgently needed.

**Objective**. The purpose of this study was to explore the time-dependent variability of emotions, physical complaints, intention, and self-efficacy in older adults (65+) using Ecological Momentary Assessment (EMA).

**Methods**. Observational data were collected in 64 healthy older adults (56.3% men; mean age $72.1 \pm 5.6$ years) using EMA. Participants answered questions regarding emotions (*i.e.*, cheerfulness, relaxation, enthusiasm, satisfaction, insecurity, anxiousness, irritation, feeling down), physical complaints (*i.e.*, fatigue, pain, dizziness, stiffness, shortness of breath), intention, and self-efficacy six times a day for seven consecutive days using a smartphone-based questionnaire. Generalized linear mixed models were used to assess the fluctuations of individual determinants within subjects and over days.

**Results**. A low variability is present for the negative emotions (*i.e.*, insecurity, anxiousness, irritation, feeling down) and physical complaints of dizziness and shortness of breath. The majority of the variance for relaxation, satisfaction, insecurity, anxiousness, irritation, feeling down, fatigue, dizziness, intention, and self-efficacy is explained by the within subjects and within days variance (42.9% to 65.8%). Hence, these determinants mainly differed within the same subject and within the same day. The between subjects variance explained the majority of the variance for cheerfulness, enthusiasm, pain, stiffness, and shortness of breath (50.2% to 67.3%). Hence, these determinants mainly differed between different subjects.

**Conclusions**. This study reveals that multiple individual-level determinants are time-dependent, and are better considered as 'dynamic' or unstable behavior determinants. This study provides us with important insights concerning the development of dynamic health behavior change interventions, anticipating real-time dynamics of determinants instead of considering determinants as stable within individuals.

## INTRODUCTION

Health behaviors such as performing regular physical activity, limiting sedentary behavior, having sufficient sleep, eating healthy food, have benefits for physical, mental, and social health (*Knowler et al., 2002*; *Dunstan et al., 2010*; *Luyster et al., 2012*; *Mammen & Faulkner, 2013*; *de Ridder et al., 2017*). Nonetheless, despite the general awareness of the beneficial health effects of these behaviors, individuals do not always perform them. In order to engage individuals more in performing healthy behaviors, health behavior change interventions have been developed targeting determinants of these healthy behaviors.

Many theoretical frameworks have been used in order to understand health behaviors and its determinants better (*Conner & Norman, 2017*). Up until now, most of the individual-level determinants (*e.g.*, intention, emotions, and self-efficacy) within these frameworks have been measured as if they are relatively stable over time (*e.g.*, assessed in a questionnaire where individuals report once on their levels of self-efficacy or intentions). Consequently, current health behavior change interventions that are developed based on the assumption that determinants are stable, are rather 'static' interventions and often lack effectiveness (*Spence & Lee, 2003*; *Shrestha et al., 2019*). Even though they are often presented as 'tailored' interventions, especially in the domain of electronic health (eHealth) and mobile health (mHealth) research (*Moss, Süle & Kohl, 2019*), the tailoring in most of these health behavior change interventions remains quite static. The intervention content is often only tailored to the baseline levels of the health behavior (*e.g.*, physical activity), or to the baseline levels of individual-level determinants by dividing individuals into subgroups (*e.g.*, low intention and high intention) based on these values. Consequently, a slightly different intervention content is provided for each subgroup. However, these interventions do not take into account the time-dependent variations of determinants, which might explain why the current 'tailored' interventions still have limited long-term effectiveness and often small effect sizes (*Shrestha et al., 2019*). In real-life, individual-level determinants probably vary within and between individuals, but also within and between days (*e.g.*, emotions). Recently, the dynamic aspect of determinants in research has come to the fore (*Conroy et al., 2011*; *Pickering et al., 2016*; *Dunton, 2017*). Previous research has indicated that explicitly taking dynamic frameworks into account might help to better understand and promote healthy behaviors, such as physical activity (*Rhodes, McEwan & Rebar, 2019*). In short, in order to match the time-dependent needs, preferences and behaviors of an individual, interventions should be dynamic (*Beckjord & Shiffman, 2014*).

Several theoretical frameworks have the potential to be used as a more dynamic framework (*e.g.*, Health Action Process Approach (*Schwarzer, 2008*), Theory of Planned Behavior (*Ajzen, 1991*), attitude, social influence and self-efficacy (ASE) model (*DeVries et al., 1998*)), but this potential is often not yet fully utilized in current research. One of those frameworks is the capability, opportunity, motivation, and behavior (COM-B) framework. The COM-B framework is an all-encompassing framework including four different constructs: capability, opportunity, motivation, and behavior itself (*Michie, van Stralen & West, 2011*). Capability is described as the psychological (*e.g.*, knowledge) and physical capacity (*e.g.*, skills and limitations) of the individual to engage in behavior. Opportunity is described as all factors outside the individual, *i.e.*, social factors (*e.g.*, social influences) as well as physical factors (*e.g.*, environmental context and resources), that make the behavior possible or trigger the behavior. Motivation is defined as all processes that energize and direct behavior, which can be reflective (*e.g.*, analytical decision-making) or automatic (*e.g.*, habitual processes and emotional responding). Capability and opportunity can both influence motivation. Additionally, capability, motivation, and opportunity influence behavior, but performing a certain behavior (*e.g.*, physical activity, sedentary behavior, dietary habits) can also alter these three constructs (*Michie, van Stralen & West, 2011*). During the day, individuals are faced with a variety of decisions, stressors, emotions, contextual influences, physical limitations or complaints, etc., that might influence their behavior. By considering the COM-B framework as a dynamic framework, it is possible to take these time-dependent variations of determinants into account. This, in turn, makes it possible to personalize health behavior interventions in a more dynamic manner. What works for one individual at a certain point in time does not necessarily work for that same individual at another time point, or even for another individual, since several factors influence behavior throughout the day, and behavior differs as a function of a person, context and time (*Conroy et al., 2011*; *Beckjord & Shiffman, 2014*). Despite the possibility to interpret the COM-B framework and other theoretical frameworks as a dynamic framework, current research does not take this sufficiently into account and thus misses important time-dependent information. To achieve this, the time-dependent variations of determinants should be identified in order to be able to personalize behavior change interventions in a more dynamic manner (*Pickering et al., 2016*; *Dunton, 2017*).

Ecological Momentary Assessment (EMA) is a method that enables capturing the time-dependent variations of behavior and its determinants. It is based on repeatedly collected real-time data on subjects' behaviors and/or experiences in their natural environments (*Shiffman, Stone & Hufford, 2008*). Previous studies already examined the time-dependent variations of some individual-level determinants by using EMA in the field of psychology (*Dockray et al., 2010*; *Pe & Kuppens, 2012*; *Houtveen & Sorbi, 2013*; *Maher et al., 2016*; *Pickering et al., 2016*; *May et al., 2018*), but evidence in the research field of physical activity and nutrition is limited. Research in university students showed that emotions (*i.e.*, motivation - automatic) change from day to day as well as within days (*Pe & Kuppens, 2012*). In older adults intra-individual variation was found in affect (*Kanning, Ebner-Priemer & Schlicht, 2015*) and within-subject variations in mood were observed in adults (*Giurgiu et al., 2019*). Physical complaints (*i.e.*, capability–physical) such as pain and

fatigue have been shown to vary throughout the day in adults (*Dockray et al., 2010*; *May et al., 2018*). Furthermore, in adults and older adults with migraine, stiffness and fatigue were found to vary throughout the day (*Houtveen & Sorbi, 2013*). Variations over time were also found for behavioral cognitions such as intention and self-efficacy (*i.e.*, motivation - reflective) (*Maher et al., 2016*; *Pickering et al., 2016*). Two previous studies in adults showed that intention and self-efficacy vary between and within days (*Maher et al., 2016*; *Pickering et al., 2016*).

In conclusion, in current physical activity research, evidence on the dynamic aspect of determinants is largely uninvestigated. In order to develop more dynamic health behavior change interventions to promote healthy behaviors (*e.g.*, physical activity, healthy dietary habits), information on the time-dependent variations of determinants is needed. Therefore, the aim of this study was to explore the time-dependent variability of several determinants of the COM-B framework, *i.e.*, emotions, physical complaints, intention, and self-efficacy, using EMA. We hypothesized that the within-subject variability would be larger in emotions and physical complaints compared to the between-subject variability. Emotions are classified as constructs with a more "automatic" nature in the COM-B framework. Such automatic behaviors or reactions to cues happen more subconscious and might therefore be subject to more within-subject variability. Previous research already showed that physical complaints can vary throughout the day (*Dockray et al., 2010*; *May et al., 2018*). In this study, we expect less within-subject variation in the cognitive constructs intention and self-efficacy, since both are classified as reflective in the COM-B framework and therefore, might happen more consciously and planned. The target population for the current study are older adults. To our knowledge, only a few previous studies examined the time-dependent variations of individual-level determinants in older adults, but the time-dependent variability was often discussed only briefly. As chronic diseases are more prevalent among older adults, and the global population of older adults is rising, which consequently implies enormous health care costs (*Rechel et al., 2009*), older adults are a crucial age group to target in health interventions during this time of increased life expectancy and 'healthy ageing'. Furthermore, since most older adults are retired and therefore, might have a more limited number of contexts in which they interact and presumably have a more flexible day schedule than individuals from other age groups, they are a very appropriate target group to receive 'in-the-moment' interventions.

## MATERIALS & METHODS

### Participants

Older adults (65+) were recruited between November 2019 and March 2020 using convenience sampling. Exclusion criteria were: (a) impaired cognition (*i.e.*, diagnosed with dementia, Alzheimer or other cognitive diseases), (b) severe impairment of vision and/or hearing, (c) not being able to walk 100 m, stand or sit independently, (d) impairment of fine motor skills, and (e) insufficient knowledge of the Dutch language.

## Procedure

Participants were told to take part in an observational study that explores the time-dependent variations in emotions, physical complaints, intention and self-efficacy, and whether these determinants are related to physical activity. All participants were visited at home twice. During the first home visit, the informed consent was signed, participants' sociodemographics were collected, and instructions for the measurement period were given. The participants received a small training on how to use the EMA-application by using printed screenshots of the application. The measurement period started the day after the first home visit and consisted out of one monitoring period which lasted seven consecutive days, five weekdays and two weekend days, in which the EMA questionnaire was triggered on the participants' smartphone six times a day (*i.e.*, the participants were required to answer the questionnaire 42-times in total). Previous research suggested that sufficient assessments per day are required to capture dynamic within-subject processes such as affective and cognitive factors which are related to physical activity (*Degroote et al., 2020*). A mean of five assessments per day (*Liao et al., 2016*) and seven assessments per day (*Degroote et al., 2020*) have been reported in previous reviews of EMA studies. Therefore, six triggers per day were given in the current study. Participants were asked to use their own smartphone during the measurement period (the lowest operating systems used were Android 5.0 and iOS 12.4), but participants who did not own a smartphone were provided with a Wiko Lenny 3 smartphone (Android 6.0). The Smartphone Ecological Momentary Assessment[3] (SEMA[3]) application (*Koval et al., 2019*) was installed on every smartphone and used to trigger the EMA questionnaire. SEMA[3] is a suite of software primarily designed for EMA that can be used on iOS and Android smartphones. To ensure adequate spacing across the day, six timeframes, each of one hour, were constructed between 9 AM and 10 PM in which one trigger was randomly given. An auditory signal prompted participants to complete the EMA questionnaire at each of the designated times. When the participant did not respond to the initial trigger, two reminders were given after approximately five minutes and ten minutes, and after 20 min, the questionnaire became unavailable until the next scheduled trigger. The measurement period was followed by a second home visit during which all measuring instruments were reassembled. Ethical approval was obtained from the Ghent University Hospital Ethics Committee before the start of the study (2019/0192).

## Measures
### Participants' characteristics

Self-reported information on gender, age, height, weight, main occupation before retirement, educational level, marital status, (grand)children, and pets were collected using a paper-based questionnaire. All items were assessed in Dutch, but the English translation is available in Questionnaire S1 and S2).

### EMA questionnaire

Experts in psychology, health sciences and EMA were involved in the development of the EMA questionnaire at multiple stages. Subsequently, cognitive interviews were executed with 10 older adults, which led to further adjustments of the questionnaire, mainly concerning the phrasing of the items (comprehensibility). In the final version of the EMA

questionnaire, eight emotions, five physical complaints, and two cognitive constructs, respectively intention and self-efficacy were each evaluated on a 7-point Likert scale. All items were assessed in Dutch, but the English translation is available in the (Questionnaire S1 and S2). The EMA questionnaire always started with questions about emotions, followed by questions about physical complaints and finally, questions assessing intention and self-efficacy. Although, the order of the questions assessing emotions could differ from questionnaire to questionnaire, since they were presented in a random order. This also applies for the questions about physical complaints. In total the EMA questionnaire consisted out of 18 questions of which 15 were discussed and used in the analyses of the current study.

*a. Emotions.* Four positive emotions (*i.e.*, cheerfulness, relaxation, enthusiasm, satisfaction) and four negative emotions (*i.e.*, insecurity, anxiousness, irritation, feeling down) were assessed. These items were selected from the items that are frequently used for the Experience Sampling Method (ESM) by the research group of Philippe Delespaul from the University of Maastricht, who also constructed these items (*Delespaul, 1995*). An example of an item is: "How cheerful were you just before you received the trigger?" Answering categories went from *e.g.*, "not at all cheerful" to "completely cheerful".

*b. Physical complaints.* Five physical complaints (*i.e.*, fatigue, pain, dizziness, stiffness, shortness of breath) were assessed that were selected from the Patient Health Questionnaire-15 (*Kroenke, Spitzer & Williams, 2002*). An example of an item is: "How fatigued did you feel just before you received the trigger?" Answer categories went from *e.g.*, "not fatigued at all" to "very fatigued".

*c. Intention and self-efficacy towards physical activity.* Intention and self-efficacy towards physical activity were also assessed in the EMA questionnaire. Intention, for example, was assessed by the item "In the next two hours, I will move for at least 10 min." The answering scale went from "strongly disagree" to "strongly agree."

## Analyses

Analyses were performed using R version 4.0.1 (*R Core Team, 2020*). Descriptive statistics were calculated for the total sample. In order to examine the time-dependent variability and to make firm conclusions about it, it is important to first identify the distribution of EMA responses. Therefore, a histogram was constructed for each determinant to visualize the distribution of the EMA responses of all participants together. In addition, for each determinant, the number of older adults who gave the same answer on the 7-point Likert scale for more than 90% of all EMA questionnaires they responded to was calculated. This was also calculated for 80% and 70%, and these numbers can be found in Supplemental Informations 4 and 5).

To explore the time-dependent variability of emotions, physical complaints, intention, and self-efficacy, generalized linear mixed models were estimated using the lme4—package (*Bates et al., 2015*). These models take into account the hierarchical data structure (*i.e.*, triggers clustered within days within individuals). Levels of between

and within variance of all emotions, physical complaints, intention, and self-efficacy were calculated by running intercept-only models (*i.e.*, models only including a fixed and two random intercepts, (one for individuals and one for days within individuals) with each of the determinants as the outcome variable. First, the between days (within subjects) variance (*i.e.*, differences within the same subject, but between different days) was obtained, by calculating the intraclass correlation coefficient (ICC) using the following formula: ICC(days) = variance(days)/((variance(subject) + variance(days) + variance(residual)). Second, the between subjects variance (*i.e.*, differences between different subjects) was calculated using the following formula: ICC(subject) = variance(subject)/((variance(subject) + variance(days) + variance(residual))and thereafter, the within days (within subject) variance (*i.e.*, differences within the same subject and within the same day) was obtained using the following formula: variance(residual)/((variance(subject) + variance(days) + variance(residual)). Finally, to visualize the time-dependent variability spaghetti plots were constructed for each determinant separately, in which the responses for each participant are represented by a different color.

## RESULTS

### Descriptive characteristics of the study sample

In total 67 older adults participated in the study. To be included in the analyses, participants had to respond to at least one third of all triggers (*i.e.*, 14 out of 42). Therefore, three participants were excluded, and analyses were performed on the remaining 64 participants. The descriptive characteristics of the participants are presented in Table 1. All participants completed the measurement period of seven days.

A total of 2,690 triggers were sent; however, 30 of them were sent inappropriately due to technical issues (*i.e.*, some participants received more than six triggers a day or some triggers were sent outside the six predefined times frames, *e.g.*, during the night) and were excluded from the analysis. Of the 2,660 triggers included in the analyses, 2,068 EMA questionnaires were completed (response rate of 77.7% and a mean of 32.3 completed questionnaires per participant), with a median response latency of 0.0 min (Q1 = 0.0 min, Q3 = 4.0 min). The median time needed to complete the EMA questionnaire was 1.8 min (Q1 = 1.4 min, Q3 = 2.3 min). Compliance rates for each participant separately were added in Supplemental Information 6. Mean values and standard deviations for all EMA items are presented in Table 2. For both the negative emotions as well as the physical complaints the mean values are rather low compared to the other individual-level determinants.

In Table 3, descriptive information regarding the response rates of participants towards EMA triggers are presented for both day of the week (*i.e.*, Monday, Tuesday, etc.) as well as the day in the measurement period (*i.e.*, first day, second day, etc.). Relatively high response rates were obtained during the whole measurement period ranging from 69.1% to 83.7%. The lowest response rates were obtained on the first two measurement days. Looking at the response rates for day of the week, the highest response rate was obtained on Tuesday and

**Table 1 Descriptive characteristics of the participants.**

| Demographics | Total sample ($n = 64$) |
|---|---|
| Men (%) | 56.3 |
| Age in years (M ± SD; range) | 72.1 ± 5.9; 65 - 86 |
| BMI (M ± SD; range) | 25.6 ± 4.1; 15.2 −36.0 |
| Non-tertiary education (%) | 57.9 |
| Main occupation before retirement | |
|     Household (%) | 4.7 |
|     Blue collar worker (%) | 35.9 |
|     White collar worker (%) | 53.2 |
|     Other (%) | 6.3 |
| Marital status | |
|     Single (%) | 3.1 |
|     Married or living together (%) | 84.4 |
|     Divorced (%) | 3.1 |
|     Widow/widower (%) | 9.4 |
| Having children (%) | 93.7 |
| Having grandchildren (%) | 89.1 |
| Having pets (%) | 23.4 |
|     Having a dog (%) | 12.6 |

**Notes.**
BMI, Body Mass Index (kg/m$^2$).

**Table 2 Mean values, standard deviations, skewness and kurtosis for all EMA items.**

| EMA data | M ± SD | Skewness | Kurtosis |
|---|---|---|---|
| Emotions | | | |
|     Cheerfulness | 4.5 ± 1.4 | −0.5 | 0.1 |
|     Relaxation | 4.7 ± 1.5 | −0.6 | 0.2 |
|     Enthusiasm | 4.2 ± 1.5 | −0.5 | −0.3 |
|     Satisfaction | 4.8 ± 1.4 | −0.6 | 0.5 |
|     Insecurity | 1.4 ± 0.9 | 3.3 | 12.5 |
|     Anxiousness | 1.2 ± 0.7 | 5.4 | 35.3 |
|     Irritation | 1.5 ± 1.0 | 2.5 | 6.9 |
|     Feeling down | 1.3 ± 0.8 | 3.6 | 15.3 |
| Physical complaints | | | |
|     Fatigue | 1.9 ± 1.1 | 1.3 | 1.8 |
|     Pain | 1.9 ± 1.1 | 1.1 | 1.2 |
|     Dizziness | 1.1 ± 0.5 | 5.0 | 31.8 |
|     Stiffness | 2.0 ± 1.1 | 1.1 | 1.6 |
|     Shortness of breath | 1.3 ± 0.7 | 3.5 | 14.0 |
| Intention | 3.7 ± 2.1 | 0.2 | −1.2 |
| Self-efficacy | 4.2 ± 2.1 | −0.1 | −1.2 |

**Notes.**
All items have a minimum of 1 and a maximum of 7, which means that they also have the same range going from 1 to 7 where 1 = not at all and 7 = very cheerful/relaxed/etc.

**Table 3   Response rates per day of the week and day in the measurement period.**

| Day of the week | Response rate (%) | Day in the measurement period | Response rate (%) |
|---|---|---|---|
| *Monday* | 78.3 | *First day* | 69.1 |
| *Tuesday* | 81.3 | *Second day* | 73.5 |
| *Wednesday* | 74.2 | *Third day* | 79.7 |
| *Thursday* | 80.6 | *Fourth day* | 78.9 |
| *Friday* | 79.9 | *Fifth day* | 83.7 |
| *Saturday* | 75.8 | *Sixth day* | 82.0 |
| *Sunday* | 71.9 | *Seventh day* | 74.2 |

the lowest response rate on Sunday, respectively 81.3% and 71.9%. Slightly lower response rates were obtained on weekend days and on Wednesdays.

For each determinant, a histogram is presented in Fig. 1 to visualize the distribution of the EMA responses of all participants together. To visualize the variability of emotions, physical complaints, intention, and self-efficacy spaghetti plots were added in Figs. S6–S20.

Table 4 provides per determinant the number of older adults who gave the same answer on the 7-point Likert scale for more than 90% of all EMA questionnaires they responded to. Giving the same answer for more than 90% of the time can be seen as a rather conservative definition of insufficient variation (*Röcke, Li & Smith, 2009*). This amount of variation is examined as a first step in order to make firm conclusions about the time-dependent variability of the determinants.

As shown in Table 4 there is a low variability for the negative emotions (*i.e.*, insecurity, anxiousness, irritation, feeling down) compared to the positive emotions (*i.e.*, cheerfulness, relaxation, enthusiasm, satisfaction). Furthermore, also for the physical complaints dizziness and shortness of breath a low variability is present compared to the other physical complaints (*i.e.*, fatigue, pain, stiffness). For the constructs intention and self-efficacy a sufficient level of variability is present.

## Variability of emotions

The variability of each emotion separately is presented in Table 5. For the positive emotions relaxation and satisfaction, the majority of the variance, 53.9% and 53.3% respectively, is explained by differences within subjects and within days. The remaining 40.1% and 40.1% of the variance is explained by differences between subjects, and 5.9% and 6.5% by differences between days (within subjects). For the positive emotions cheerfulness and enthusiasm 42.4% and 40.6% of the variance is explained by differences within subjects and within days. The majority of the variance in cheerfulness and enthusiasm, 50.2% and 53.3% respectively, is explained by differences between subjects. Differences between days (within subjects) explain 7.4% and 6.1% of the variance.

For the negative emotions, 42.9% to 56.4% of the variance in the emotions insecurity, anxiousness, irritation, and feeling down is explained by differences within subjects and
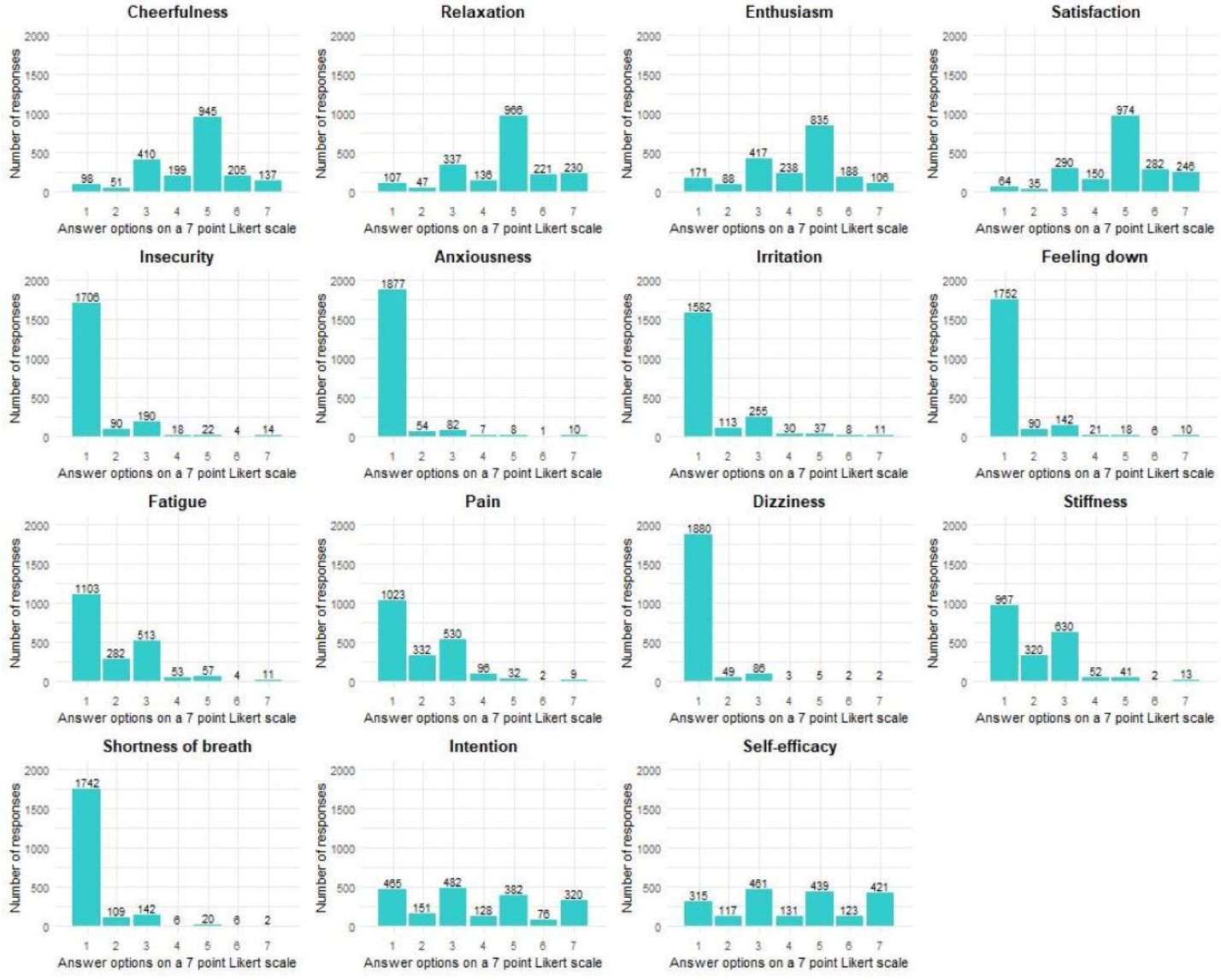

**Figure 1  Distribution of the responses per determinant for all participants together.**

within days. Differences between subjects explain 29.2% to 43.4% of the variance, and differences between days (within subjects) 7.9% to 18.5%.

## Variability of physical complaints

All values for each physical complaint separately are presented in Table 5. Only 22.3% to 29.8% of the variance in the physical complaints pain, stiffness, and shortness of breath is explained by differences within subjects and within days, whereas 54.8% to 67.3% is explained by differences between subjects. Differences between days (within subjects) explain 10.4% to 17.1% of the variance. For fatigue and dizziness the majority of the

**Table 4** Percentage of participants who gave the same answer to the EMA question for more than 90% of the triggers.

| Emotions | Participants who gave the same answer for more than 90% of the triggers | | Physical complaints | Participants who gave the same answer for more than 90% of the triggers | |
|---|---|---|---|---|---|
| | N | % | | N | % |
| *Cheerfulness* | 6 | 9.4 | *Fatigue* | 14 | 21.9 |
| *Relaxation* | 3 | 4.7 | *Pain* | 23 | 35.9 |
| *Enthusiasm* | 7 | 10.9 | *Dizziness* | 53 | 82.8 |
| *Satisfaction* | 5 | 7.8 | *Stiffness* | 16 | 25.0 |
| *Insecurity* | 36 | 56.3 | *Shortness of breath* | 45 | 70.3 |
| *Anxiousness* | 48 | 75.0 | **Intention and self-efficacy** | **N** | **%** |
| *Irritation* | 24 | 37.5 | *Intention* | 3 | 4.7 |
| *Feeling down* | 29 | 45.3 | *Self-efficacy* | 4 | 6.3 |

**Table 5** Between and within variance in percent for each determinant.

| | Between days (within subjects) variance (%) | Between subjects variance (%) | Within subjects and within days variance (%) |
|---|---|---|---|
| *Emotions* | | | |
| Cheerfulness | 7.4 | 50.2 | 42.4 |
| Relaxation | 5.9 | 40.1 | 53.9 |
| Enthusiasm | 6.1 | 53.3 | 40.6 |
| Satisfaction | 6.5 | 40.1 | 53.3 |
| Insecurity | 7.9 | 43.4 | 48.7 |
| Anxiousness | 18.5 | 38.6 | 42.9 |
| Irritation | 14.4 | 29.2 | 56.4 |
| Feeling down | 14.3 | 34.6 | 51.1 |
| *Physical complaints* | | | |
| Fatigue | 13.7 | 34.4 | 51.9 |
| Pain | 10.4 | 67.3 | 22.3 |
| Dizziness | 14.2 | 38.6 | 47.2 |
| Stiffness | 17.1 | 54.8 | 28.1 |
| Shortness of breath | 12.4 | 57.9 | 29.8 |
| *Intention and self-efficacy towards physical activity* | | | |
| Intention | 2.0 | 32.2 | 65.8 |
| Self-efficacy | 6.2 | 41.3 | 52.5 |

variance, 51.9% and 47.2% respectively, is explained by differences within subjects and within days.

## Variability of intention and self-efficacy towards physical activity

Values for intention and self-efficacy separately are available in Table 5. Differences within subjects and within days explain 65.8% and 52.5% of the variance for intention and self-efficacy. Furthermore, 32.2% and 41.3% of the variance is explained by differences
between subjects, and differences between days (within subjects) explain 2.0% and 6.2% of the variance.

## DISCUSSION

The aim of the current study was to explore the time-dependent variability of emotions, physical complaints, intention, and self-efficacy, using ecological momentary assessment in healthy older adults. In general, two important conclusions can be drawn from the results. The first conclusion is that one should examine first whether specific individual-level determinants are present in a specific study sample, before deciding on whether or not to focus on these determinants in further personalized interventions. In the current study, higher mean values were obtained for the positive emotions compared to negative emotions, which is in line with previous research in older adults (*Mather & Carstensen, 2005*), and relatively low mean values were obtained for all physical complaints. Hence, this sample of older adults experienced fewer negative emotions and did not experience many physical complaints during the measurement period. Consequently, not all determinants examined in the current study were equally present. In addition, the majority of participants reported the same answer for more than 90% of the EMA questionnaires they responded to for the negative emotions anxiousness (*i.e.*, not anxious at all) and insecurity (*i.e.*, not insecure at all), and the physical complaints of dizziness (*i.e.*, not dizzy at all) and shortness of breath (*i.e.*, not short of breath at all). Therefore, it will be unlikely to find evidence for time-dependent variations within subjects within days and more likely to find evidence for variations between different subjects since only a minority of the participants experienced *e.g.*, dizziness. These findings indicate that several determinants (*i.e.*, anxiousness, insecurity, dizziness and shortness of breath) are not highly present in the current sample of older adults and therefore, might even vary insufficiently. Previous research (*Röcke, Li & Smith, 2009*; *Gruber et al., 2013*; *Pickering et al., 2016*) often assumed that all determinants were equally present in the sample of interest, which might lead to wrong conclusions. Therefore, future research should first look at the presence of individual-level determinants and examine whether a sufficient variation is present, before looking at the time-dependent variability. In this regard, it would be of great value to have a cutoff value to determine whether a determinant shows sufficient variation to further examine the time-dependent variability of determinants. For example, when more than 75% of the participants give the same answer for more than 90% of all triggers they responded to, we can conclude that this determinant has a low and maybe even insufficient variation and therefore, it would be unlikely to find evidence for within subject variations. Consequently, it might not be meaningful to further explore the time-dependent variability of those determinants from a content point of view. For future health behavior change interventions, it is interesting to know which determinants are time-dependent and therefore, should be focused on as dynamic. More research is needed in order to create such a cut off value.

The second general conclusion is that determinants can vary over time. In the current study was found that the majority of the variance of various determinants (*i.e.*, relaxation,

satisfaction, insecurity, anxiousness, irritation, feeling down, fatigue, dizziness, intention, and self-efficacy) can be explained by the within subject variance, indicating a variation over time within the same person. For example, someone can be very relaxed in the morning, but during the afternoon this relaxation decreases and as the day further progresses the same person might feel more relaxed again in the evening. These time-dependent variations should be targeted in future health behavior interventions in order to personalize interventions in a more dynamic matter (*Dunton & Atienza, 2009*). Suggestions to be active can be adjusted to the cognitive, emotional and physical 'state' of the individual at that moment in order to change the behavior of interest. For example, smartphone notifications containing suggestions such as going for a short walk or run can be sent when the determinants are 'right' (*e.g.*, high intention to be active or absence of physical complaints). Just-in-time adaptive interventions provide the opportunity to interact at the 'right' time and in the 'right' context in order to change behavior more effectively (*Nahum-Shani et al., 2018*). To determine this 'right' time, profiling for each individual separately might be needed at the start of a dynamic intervention. For example, by asking participants to fill in questions about individual-level determinants for a few days in order to map individual fluctuations, and to know when the time is 'right' to trigger this specific individual. In the current study was also found that for a few determinants (*i.e.*, cheerfulness, enthusiasm, pain, stiffness, and shortness of breath), the majority of the variance is explained by differences between subjects. For example, one person can be very cheerful, whereas another person is not. Although, it should be taken into account that the mean values for pain, stiffness, and shortness of breath were generally low. This finding argues in favor of personalizing interventions more at the individual level, as one size fits all interventions may not be equally effective in different people (*Spence & Lee, 2003*). Lastly, the current study showed that only a small part of the variance can be explained by the between days variance, which indicates that the variations within the same subject but between different days are limited. This finding might apply specifically to older adults since most of them are retired and have a similar daily structure for multiple days, causing limited variations between days. The limited between days variation is an interesting finding regarding profiling in health behavior interventions, since collecting data for only a few days before the start of a personalized intervention might be enough to have sufficient information about the individual fluctuations of determinants in older adults. Our hypothesis for the current study was that the within-subject variability would be larger in emotions and physical complaints, while less within-subject variation was expected in the cognitive constructs intention and self-efficacy. The results of the current study show that multiple emotions and physical complaints have more within-subject variation, and in contrast to our hypothesis, also intention and self-efficacy vary more within-subjects then between-subjects.

Identifying the time-dependent and context-dependent variations is a first step to develop dynamic health behavior interventions and, more specifically, just-in-time adaptive interventions. By using just-in-time adaptive interventions older adults can be stimulated to perform healthy behaviors (*e.g.*, be physically active, limit sedentary behavior, eating healthy) when the time is 'right' (*i.e.*, when individuals are receptive and vulnerable)

and the context is 'right' (*e.g.*, when they are close to a park) (*Nahum-Shani et al., 2018*). Future research should explore the variability of other determinants in older adults to gain more insight into the time-dependent, but also context-dependent variations of determinants (*i.e.*, by asking specific context-related questions such as "Are you outside or inside?" or "Is there a park in your direct environment?"). In order to determine which determinants should be examined, theoretical frameworks might be helpful if they are interpreted as dynamic frameworks. Specifically, it might be interesting to further explore the time-dependency of other determinants in the COM-B framework such as memory, attention and decision processes (*i.e.*, capability –psychological), behavioral regulation (*i.e.*, capability –psychological) or social influences (*i.e.*, opportunity –social), or determinants in other theoretical frameworks. In the current study, we limited the number of questions so that the EMA questionnaire could be completed in only a few minutes which minimized the burden on the participants (*Degroote et al., 2020*). Therefore, we could not include all determinants mentioned in the COM-B framework, but selected a few (*i.e.*, emotions, physical complaints, intention, and self-efficacy). The identification of the time-dependent variability of determinants is not only important for physical activity, but is also very relevant for other health behaviors such as sedentary behavior or healthy eating habits. Future health behavior change interventions focusing on other healthy behaviors can then also take these variations into account and further improve their health effects.

The current study demonstrates the importance and added value of using EMA. This assessment strategy allows researchers to identify time-dependent as well as context-dependent variations, while traditional questionnaires come with retrospective biases (*Dunton & Atienza, 2009*). However, not many EMA studies have been conducted in older adults. Researchers might have the impression that performing an EMA study using mobile technology in older adults is not feasible. Nonetheless, previous studies already proved this assumption wrong (*King et al., 2013*; *Ramsey et al., 2016*). In the current study relatively high response rates were obtained for the whole measurement period ranging from 69.1% to 83.7%. Previous research found average response rates of 77% in adults (*Hofmann & Patel, 2015*) and 84% in both adults and older adults (*Thai & Page-Gould, 2018*). In the current study, the lowest response rates were obtained on the first two measurement days, which might indicate that participants needed some time to get used to the smartphone application. Therefore, in future studies, a more extensive training on how to use specific smartphone applications is recommended to increase the response rate even more. In the current study, the highest response rate was obtained on Tuesday and the lowest response rate on Sunday. The response rates on weekend days and on Wednesdays are slightly lower than the response rates on the other weekdays. It might be possible that the participants planned more activities during the weekend (*e.g.*, with family) and on Wednesday (*e.g.*, with grandchildren) and, therefore, they might have been less focused on filling in the EMA questionnaires. However, these findings were not verified statistically. Interestingly, participants without prior experience with smartphone use were also able to complete the EMA questionnaires. However, it is possible that they filled out fewer EMA questionnaires than participants with more smartphone experience. Nevertheless, these findings highlight the feasibility of conducting EMA studies using mobile applications among older adults.

## Strengths and limitations

This study has several strengths. First, in the research field of physical activity, this is one of the first studies using EMA in older adults. Second, time-dependent variations of determinants are largely uninvestigated and therefore, the current study provides valuable insights into the time-dependent variability of several determinants (*i.e.*, emotions, physical complaints, intention, and self-efficacy) in older adults. Consequently, this study fills an important gap and is a valuable contribution to the current literature. Third, as we have also included older adults who did not own a smartphone and had no previous smartphone experience, the study results can be generalized to populations with low technological literacy. Fourth, physical complaints are easily overlooked in studies on physical activity in older adults, but these were included in the current study. Finally, in EMA shorter time frames are used to assess the variables of interest, which minimizes recall bias and might be especially relevant for older adults (*Knell et al., 2017*).

Our study also has some limitations. The first drawback is the fact that convenience sampling was used to recruit participants. In this regard, we have to be careful with the generalization of the results to a wider population of older adults. In future studies it is recommended to use a random sample. Second, it is possible that some participants were less familiar with using a smartphone and therefore, filled out fewer EMA questionnaires. Consequently, we might have missed some important time-dependent information. For future studies, it is recommended to assess the smartphone experience of older adults and, if necessary, to provide a training session to learn how to use the smartphone application that is used in the study. Third, the findings of the current study are specific to the sample of older adults used in the current study. Overall low mean values were obtained for the negative emotions and the physical complaints, whereas higher mean values for negative emotions and/or physical complaints might be obtained in another sample of older adults and consequently, other findings might be observed. Therefore, profiling is recommended before the start of an intervention to get to known the specific sample of interest. Fourth, since the minimum age was 65 years and the oldest participant was 86 years, it might be possible that we recruited a rather heterogeneous sample of older adults. For future studies, it is recommended to use a smaller age range in order to obtain a more homogeneous sample of older adults. Fifth, the items used in the EMA questionnaire were not specifically validated for EMA research or for the specific target group of the current study. For future EMA studies, it is recommended to use items that are validated for EMA and if possible, also for the target group. Sixth, it is possible that the answers on the EMA questionnaire were affected by previous answers. This should be taken into account in the analysis of future EMA research.

## CONCLUSIONS

In conclusion, emotions (*i.e.*, relaxation, satisfaction, insecurity, anxiousness, irritation, feeling down), physical complaints (*i.e.*, fatigue, dizziness), intention, and self-efficacy can vary over time in older adults. Consequently, these determinants are time-dependent, and therefore should be treated as 'dynamic' behavior determinants. This implies that

more dynamic health interventions are needed. It is important to keep in mind that the presence of the determinants should be examined first before looking at the time-dependent variability of the determinants.

This study provides important insights concerning the development of dynamic health behavior change interventions, anticipating real-time dynamics of determinants instead of considering determinants as stable within individuals.

## ACKNOWLEDGEMENTS

The authors would like to thank the master students and intern for their contribution in data collection, and all older adults for their participation in the study.

### Funding
Lieze Mertens (FWO17/PDO/140) is supported by a postdoctoral fellowship of the Research Foundation Flanders. The funders had no role in study design, data collection and analysis, decision to publish, or preparation of the manuscript.

### Grant Disclosures
The following grant information was disclosed by the authors:
Research Foundation Flanders: FWO17/PDO/140.

### Competing Interests
The authors declare there are no competing interests.

### Author Contributions

- Iris Maes conceived and designed the experiments, performed the experiments, analyzed the data, prepared figures and/or tables, authored or reviewed drafts of the paper, and approved the final draft.
- Lieze Mertens conceived and designed the experiments, analyzed the data, prepared figures and/or tables, authored or reviewed drafts of the paper, and approved the final draft.
- Louise Poppe conceived and designed the experiments, analyzed the data, authored or reviewed drafts of the paper, and approved the final draft.
- Geert Crombez conceived and designed the experiments, authored or reviewed drafts of the paper, and approved the final draft.
- Tomas Vetrovsky conceived and designed the experiments, authored or reviewed drafts of the paper, and approved the final draft.
- Delfien Van Dyck conceived and designed the experiments, prepared figures and/or tables, authored or reviewed drafts of the paper, and approved the final draft.

### Human Ethics
The following information was supplied relating to ethical approvals ({i.e.}, approving body and any reference numbers):

Ethical approval was obtained from the Ghent University Hospital Ethics Committee before the start of the study (2019/0192).

## Data Availability

The raw data and the R code are available in the Supplementary File.

## Supplemental Information

Supplemental information for this article can be found online at http://dx.doi.org/10.7717/peerj.13234#supplemental-information.

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
