# Peer review of "The variability of emotions, physical complaints, intention, and self-efficacy: an ecological momentary assessment study in older adults"

_PeerJ, doi:10.7717/peerj.13234_

## Round 0.1 · original submission · Major Revisions

Dear authors:

We have now received three reviews of your manuscript and we would like to urge you to revise and resubmit.

The first and second reviewers have reservations about the rationale and design of your study, as well as some methodological concerns; reviewer 3 raises more minor points.

If you decide to revise and resubmit - please do so and document all changes in your rebuttal letter in order of the points raised by each of the three reviewers.

Reviewer 1 ·

Basic reporting

Clear English is used.
No background or justification is provided for studying older adults (besides the claim that is was not performed before). This population has various characteristics that should be described, relevant to the intervention in test here. Motivation is widely studied in older adults, as well as effects of physical activity on pain, cognition and emotions. Even methodologically, older adults are the age group with highest within-participant variability (that is seen in the results) and should have been expected. The introduction is missing an important part, as I see it.
What were the study hypotheses? This part is missing.
Please find more comments in the attached file.

Experimental design

I was surprised to find out that no information regarding physical activity was collected, hence correlation with measured variables did not receive validation of relevancy and importance. Eventually, I remained with these questions: Is it important to distinguish stable and non-stable emotions? Along this categorization, which are relevant to health behavior and how? What are the practical implications of the results? I can see the potential of contribution but I feel that the problem should have been better explained, the population should have received relevant elaboration in the introduction and discussion and the implications should have been more clearly stated.
Please find more comments in the attached file.

Validity of the findings

Since there were no hypotheses, I'm afraid I was not able to evaluate the results section properly.
only descriptive statistics were discussed.

Annotated reviews are not available for download in order to protect the identity of reviewers who chose to remain anonymous.

Reviewer 2 ·

Basic reporting

I would like to thank the authors for presenting this interesting paper on a highly relevant and timely research question. I enjoyed reading the paper which is written in clear English.

Regarding the "Basic Reporting", I would like to bring up two major points that would improve the manuscript very significantly:

a) The authors do not cover the recent literature in this field. For example, solely the within-subject association of mood/affective states/well-being and physical activity and its relevance for health behavior change has been researched by numerous ambulatory assessment studies (including the elderly as a target group, see e.g.: https://scholar.google.com/scholar?hl=de&as_sdt=0%2C5&q=physical+activity%2C+mood%2C+ambulatory+assessment&btnG=) - here I suggest the authors to include more of this prior works and their findings into their introduction and discussion section. For example line 126 "no previous studies examined" => see, e.g., https://ijbnpa.biomedcentral.com/articles/10.1186/s12966-015-0272-7

b) The authors claim to research "The variability [...] TOWARDS physical activity", however, they do not include physical activity as a outcome variable of interest into their multi-level models. They "only" investigate the variabilty of these constructs on different resolutions. While the latter one is certainly the first step when reseraching "The variability [...] TOWARDS physical activity" it remains to be discussed whether this is enough to state the term "TOWARDS physical activity". So, if the authors did assess data on physical activity within their study, I would strongly encourage them to include models with PA as outcome being predicted by "emotions, physical complaints, intention, and self efficacy" into their manuscript. This would highly benfit their research approach. If these kind of data had not been captured, I would suggest to tone down the argumentation "TOWARDS physical activity" throughout the paper since this is then "only" based on theoretical assumptions not being supported by the evidence provided in the paper itself. This also concerns naming the variables "determinants" throughout the manuscript.

Experimental design

Table 2: The table would benefit from additional data presentation, e.g., skewness, kurtosis, minimum, maximun (range) etc.

Table 3: Which differences of weekdays are statistically significant?

Table 4: Presenting data for other cut-offs would be helpful, you could also include histograms here

Did you think about conducting a multi-level factor analysis on this data? This might unravel more in depth insights on how this data is clustered.

Figure 1: Since within-person changes are especially important in relation to PA, you could additionally plot this data after within-subject centering of the variables

Validity of the findings

Discussion, line 319: The authors suggest research to create a cut-off value. In multi-level modelling, even a low within-subject variance is seen as sufficient to model within-subject associations. On which basis would you built such a cut-off value, how would you derive it? A broder discussion both from a methodolocial/statistical and from the health behavior / content-related side building upon the current state of the art literature would be helpful here.

Line 376: For the compliance reported authors could refer to current ambulatory assessment guidelines to set them into context: https://scholar.google.com/scholar?hl=de&as_sdt=0%2C5&q=ambulatory+assessment+guidelines&btnG=.

In general, the discussion would benefit from a restructuring and a more concise writing.

Additional comments

Minor points:

a) It would be helpful for readers to have statistics included to the results section of the abstract.
b) line 294 "fewer" => was it fewer or less strong emotions: qualtiy vs. quantity
c) overall, the discussion lacks citations for many of the topics discussed, e.g., for geolocation tracking (line 358): https://scholar.google.com/scholar?hl=de&as_sdt=0%2C5&q=physical+activity%2C+mood%2C+ambulatory+assessment%2C+geolocation+tracking&btnG=&oq=physical+activity%2C+mood%2C+ambulatory+assessment%2C+geolocation+track

x) Suggestions for language improvements:
"in reality" => in real-life
"time-based EMA" => EMA
line 343 "will" => is this evidenced? Here "may" could be a more appropriate term
line 370 "cannot" => better: traditional questionnaires come with retrospective biases

·

Basic reporting

Thank you for the opportunity to review the manuscript. I believe that this will be of great interest to the readers as well as researchers in the field. This well-written article focused on time-dependent variability of emotions, physical complaints, intention, and self-efficacy in older adults. Below are some comments for the authors to consider to improve the clarity of the manuscript:

1. The Introduction provides a great overview of the relevance for the need of developing further dynamic within-subject frameworks. I would like to read the author's opinion about what is necessary to create such a dynamic framework. In other words, is it possible to state in general terms what are the characteristics of a dynamic within-subject model/theory?

Experimental design

The manuscripts provide original research on a high technical and ethical standard with sufficient information for replication. Author“s may consider the following comments:

2. The authors provided almost all relevant data when using EMA. However, I suggest double-checking the STROBE Checklist for Reporting EMA studies (10.2196/jmir.4954) or further recommendations (https://doi.org/10.1037/abn0000473) to add information (e.g., compliance rate per participant).
3. Could the authors provide some information about psychometric properties? (e.g., within-subject reliability of the used items)
4. The description of the analyses could be improved. If I am right, the authors calculated the Intraclass correlation coefficient (ICC) of a 3-Level data structure by separating the variance into three parts. I would suggest presenting the formula of the calculated variances.
5. Did the authors check some model assumptions?

Validity of the findings

The findings are interesting and helpful for researchers while planing within-subject interventions. Author“s may consider the following comments:

6. I was wondering whether the time-depending determinants could also be relevant for sedentary behavior. Especially since older adults are prone to spend a lot of time sedentary, it might be worth discussing that future interventions, as well as dynamic frameworks, should also take this into account.
7. Line 380-384: I would suggest reporting no new results within the discussion section and moving this part to the results section.
8. The results about the participants who gave the same answer to the EMA question for more than 90% is really interesting and I like the discussion about a potential threshold. Did the authors check if there are potential links with participant characteristics (e.g., age, own smartphone vs. study smartphone, smartphone experiences)?

Additional comments

no comment

---

## Round 0.2 · Minor Revisions

Great job on the major revisions. Please address the remaining concerns of Reviewer 1.

Reviewer 1 ·

Basic reporting

I found great improvements in the reporting.

Experimental design

I believe the goal of the study should be more precise, stating the exploratory nature of the design.

Validity of the findings

Here again, if hypotheses are precise and no expectation is made regarding physical activity, then findings will be valid.

Additional comments

Thank you for your thorough work, the MS has definitely improved and much more suitable for publication now.
Two issues are still concerning me:
1. I still feel that the reader should be informed of the study goals and hypotheses more accurately, excluding the physical activity background (should receive less volume) and exploratory nature should be stated (since the results part is otherwise very weak).
2. Stating that chronic diseases accompany aging and that the population is aging is not enough as motivation for the study, I'm afraid. I think a more accurate reference should be made for the difficulties and potential benefit of the tested issues.

·

Basic reporting

I have no concerns. The authors addressed all issues adequately.

Experimental design

I have no concerns. The authors addressed all issues adequately.

Validity of the findings

I have no concerns. The authors addressed all issues adequately.

Additional comments

I have no concerns. The authors addressed all issues adequately.

---

## Round 0.3 · accepted · Accept

This manuscript has undergone significant revisions. Thank you for putting in the effort.